# Cation controlled rotation in anionic pillar[5]arenes and its application for fluorescence switch

Hao Zheng[1], Lulu Fu[1], Ranran Wang[1], Jianmin Jiao[1], Yingying Song[1], Conghao Shi[1], Yuan Chen[1], Juli Jiang [1,2] ✉, Chen Lin [1] ✉, Jing Ma [1] ✉ & Leyong Wang [1]

Controlling molecular motion is one of hot topics in the field of chemistry. Molecular rotors have wide applications in building nanomachines and functional materials, due to their controllable rotations. Hence, the development of novel rotor systems, controlled by external stimuli, is desirable. Pillar[n] arenes, a class of macrocycles, have a unique planar chirality, in which two stable conformational isomers $p$R and $p$S would interconvert by oxygen-through-the-annulus rotations of their hydroquinone rings. We observe the differential kinetic traits of planar chirality transformation in sodium carboxylate pillar[5]arene (**WP5-Na**) and ammonium carboxylate pillar[5] arene (**WP5-NH$_4$**), which inspire us to construct a promising rotary platform in anionic pillar[5]arenes (**WP5**) skeletons. Herein, we demonstrate the non-negligible effect of counter cations on rotational barriers of hydroquinone rings in **WP5**, which enables a cation grease/brake rotor system. Applications of this tunable rotor system as fluorescence switch and anti-counterfeiting ink are further explored.

Molecular rotors have various potential applications in molecular devices[1–3], medicine[4–6], asymmetric catalysis[7,8], and smart materials[9–11], which were derived from their controllable rotations. For example, light-driven molecular motors could drill through cell membranes using their molecular-scale actuation, inducing necrosis and facilitating chemical species into live cells[5]. Feringa and co-workers developed an artificial muscle-like functional materials which was based on supramolecular assembly of photoresponsive molecular motors[9]. Therefore, it is of great significance to develop novel rotor systems with controllable manners. As an emerging class of macrocycles, pillar[n]arenes have attracted wide attentions due to their electron-rich cavities, and played important roles in the field of supramolecular chemistry[12–16]. Pillar[n]arenes have a unique planar chirality, in which two stable conformations $p$R and $p$S could interconvert by oxygen-through-the-annulus rotations of their hydroquinone rings (Fig. 1a)[17–20]. Hence, pillar[n]arenes could be considered as promising rotor platforms.

Recently, a variety of strategies have been established to lock the planar chirality of pillar[n]arenes, such as guest[21–24], solvent[25–27], temperature[28–30], as well as redox[31]. However, these strategies were based on thermodynamic control, while the research on kinetic control of transformation was rarely reported. Steric hindrance and intramolecular hydrogen bonds have been found crucial impacts on rotational barriers of hydroquinone units in pillar[5]arenes, which were reported by Ogoshi and Stoddart, respectively[32,33]. Nevertheless, the kinetic control on rotations in pillar[n]arenes, and in particular, the behavior of switchable rotary motions of them was unexplored so far.

Sodium carboxylate pillar[5]arene (**WP5-Na**) and ammonium carboxylate pillar[5]arene (**WP5-NH$_4$**) (Fig. 1a), are widely used in aqueous supramolecular assembly systems[34–37]. Hitherto, the effect of counter cations on **WP5** had never been reported. According to the reported $^1$H NMR spectra of **WP5-Na** and **WP5-NH$_4$** in D$_2$O[34–37], we serendipitously noticed that the signal of methylene group in the rims

[1]State Key Laboratory of Analytical Chemistry for Life Science, Jiangsu Key Laboratory of Advanced Organic Materials, School of Chemistry and Chemical Engineering,  Nanjing University, 210023 Nanjing, China. [2]Ma'anShan High-Tech Research Institute of Nanjing University, Ma'anShan 238200, China. ✉e-mail: jjl@nju.edu.cn; linchen@nju.edu.cn; majing@nju.edu.cn

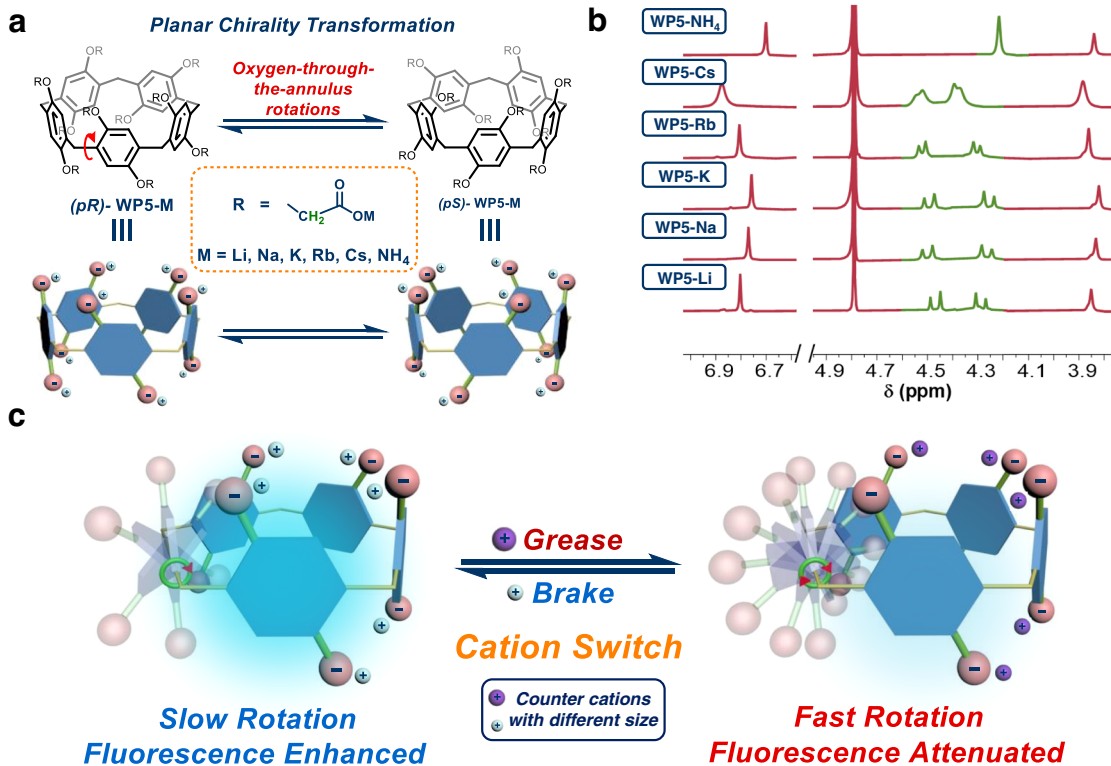

**Fig. 1 | Overview of controllable rotor platforms based on WP5 skeletons.**
**a** Illustration of planar chirality transformation of **WP5-M**. **b** [1]H NMR spectra of
**WP5-M** in $D_2O$ (10.0 mM, 298 K), peaks of methylene groups are shown in green.
**c** Cartoon Illustrations of cation-modulated switchable rotary motion of **WP5-M**
(silver and purple balls represent different counter cations with distinct size). For
simplification, only one rotary phenolic ring of **WP5** is shown.

of **WP5-Na** was split into two doublets, while in **WP5-NH₄** case it was a
singlet (marked in green, Fig. 1b), indicating slower rotations of
hydroquinone moieties in **WP5-Na** and faster rotations in **WP5-NH₄**[38].
We supposed such distinct kinetic properties may attribute to impacts
of counterions Na⁺ and NH₄⁺ on rotational barriers of hydroquinone
rings in **WP5**. This discovery provided us possibilities to construct
switchable rotations in **WP5** by means of cation switch. In addition, it is
worth noting that one of the working mechanisms of the aggregation-
induced emission (AIE) phenomena was the restriction of intramole-
cular rotations (RIR, e.g., tetraphenylethylene derivatives), of which
the aggregation could reduce the speed of intramolecular rotations,
lowering energy dissipations, and thus the fluorescence intensity
could be enhanced[11]. Moreover, pillar[n]arenes have been recently
reported as AIEgens (luminogens exhibiting AIE attributes), and their
AIE properties were arisen from restricted rotations of their phenolic
rings[39,40]. Inspired by that, we inferred controlling rotary speed of
hydroquinone rings in **WP5** would thus result in fluorescence switch.

In this work, we investigate the effect of various counterions on
rotations of hydroquinone rings in **WP5**, establishing a controllable
rotary system, and its applications in fluorescence switch and anti-
counterfeiting inks are further developed (Fig. 1c and Supplementary
Movie 1).

## Results

### Design and synthesis

To explore the impact of counterions on rotations of hydroquinone
units in anionic pillar[5]arenes, carboxylate pillar[5]arenes with various
monovalent counterions were chosen (**WP5-M**, Fig. 1a), which could be
completely ionized in water. In this work, we focused on alkali metal
ions (i.e., Li⁺, Na⁺, K⁺, Rb⁺, Cs⁺), and ammonium ion (NH₄⁺) was also
included (Fig. 1a). These anionic pillar[5]arenes were facilely synthe-
sized by reacting the carboxylic-substituted pillar[5]arene (**P5-COOH**)
with the hydroxide of corresponding cations (Fig. 2 and

Supplementary Figs. 2–11). Comparing the [1]H NMR spectra of **WP5-M**
in $D_2O$ at 298 K, it was found that methylene peaks in the rims of **WP5-
Li**, **WP5-Na**, **WP5-K**, **WP5-Rb**, and **WP5-Cs** were split into two doublets,
indicating their non-equivalency (i.e., slow rotations), while **WP5-NH₄**
exhibited a singlet peak, referring to the fastest rotation among **WP5**
derivativities (Fig. 1b).

### Variable-Temperature NMR studies

For quantitatively examining rotational barriers of rotors, **WP5-M** were
then subjected to variable temperature (VT) NMR studies[32,33,38,41,42]. The
two doublet methylene peaks tended to coalesce at 343 K, 338 K,
333 K, 333 K, and 328 K for **WP5-Li**, **WP5-Na**, **WP5-K**, **WP5-Rb**, **WP5-Cs**
in $D_2O$, respectively (Fig. 3). A rotational barrier ($\Delta G^{\ddagger}$) of 18.23, 17.55,
15.95, 15.68, and 15.06 kcal/mol was revealed based on Eyring plots[38], of
which the accuracy was validated by three independent measurements
(Table 1 and Supplementary Figs. 12–21). This result suggested that
counter cations of **WP5-M** affected rotational barriers of hydro-
quinone rings in **WP5-M** indeed. Interestingly, a linear line was
obtained when we tried to correlate the experimental rotational bar-
riers with the radius r of corresponding cations ($R^2 = 0.99$, Fig. 4a). It
decreases in rotational barrier $\Delta G^{\ddagger}$ as the ionic radius increases
(negative slope). Li⁺, with a smallest radius of 0.76 Å, gave the largest
$\Delta G^{\ddagger}$ (i.e., slowest rotation speed). Although it was a failure to measure
the rotational barrier of **WP5-NH₄** owing to the limitation of the
freezing point of deuteroxide, a qualitative result could be drawn that
**WP5-NH₄** possessed the lowest rotational barrier.

We inferred that the decreasing tendency of rotational barriers of
**WP5-M** could be ascribed to energy difference of ground states (GS) or
transition states (TS), which was induced by counter cations of **WP5-M**
(Supplementary Fig. 22). The impact of GS energy on rotational bar-
riers should be minor, since NH₄⁺ possessed stronger stability to **WP5**
anion than K⁺ (contribution of hydrogen bonding) in ground states,
but the rotational barrier of **WP5-NH₄** was much lower than that of

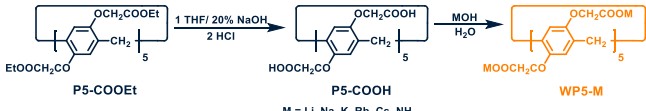

**Fig. 2 | General synthetic routes of WP5-M.** WP5-M was obtained by reacting the hydroxide of corresponding cations (MOH) with **P5-COOH**, the hydrolysis product of **P5-COOEt**.

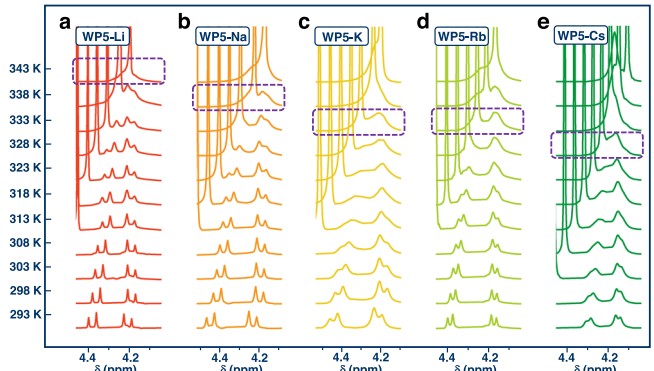

**Fig. 3 | Partial VT NMR spectra of WP5-M in D$_2$O.** The coalesce temperature for **WP5-Li** (**a**), **WP5-Na** (**b**), **WP5-K** (**c**), **WP5-Rb** (**d**), and **WP5-Cs** (**e**) was 343 K, 338 K, 333 K, 333 K, and 328 K, respectively (marked with purple dashed). The concentration of rotors was 10.0 mM.

**WP5-K** (Supplementary Fig. 23). Besides, a linear relationship between experimental $\Delta S^{\ddagger}$ and $\Delta H^{\ddagger}$ was observed, which was a typical enthalpy-entropy compensation ($R^2 = 1.00$, Fig. 4b)[43]. It suggested the existence of enhanced binding force (enthalpy favored) in TS from **WP5-Li** to **WP5-Cs**, which would lead to reduction of configurational freedom (entropy decrease). Therefore, we speculated that this downward trend of rotational barriers could be interpreted as following: in rotary process, there was a strong electronic repulsion between carboxylate anions in the rims and the electron-rich cavity of pillar[5]arenes (Fig. 4c). Cation acting as grease, could insert between anions and cavities, involving coulomb interactions with anions and cation-π interactions with cavities, by which the disfavored repulsion could be impaired. Ascribing to a size-specific mechanism that was ubiquitous in macrocyclic host-guest systems (e.g., the binding constant of 18-crown-6 with K$^+$ was higher than that with Na$^+$ and the size of K$^+$ that was more matched with the cavity of 18-crown-6)[44–46], larger counter cations might have stronger interactions, leading to an enhanced lubricant effect (Fig. 4c). Furthermore, although radii of NH$_4^+$ and K$^+$ were similar (1.43 Å and 1.38 Å, respectively), **WP5-NH$_4$** had a lowest rotational barrier, which could be attributed to additional hydrogen bond between ammonium ion and carboxylate oxygen. To preliminarily validate this size-specific mechanism in pillar[5]arenes, we measured binding constants ($K_a$) of M$^+$ with ethoxycarbonylmethoxy-substituted pillar[5]arene (**P5-COOEt**, Fig. 2) in DMF/H$_2$O solution (v/v = 4/1) using titration method (Supplementary Fig. 26)[47,48]. Although it was failed to measure $K_a$ of Li$^+$ and Na$^+$ due to the fact that the changes of their UV absorption were too small to conduct the non-linear curve-fitting, an increased $K_a$ of K$^+$, Rb$^+$, and NH$_4^+$ was successfully estimated to be 4.2 ($\pm$ 1.3) $\times 10^2$, 7.9 ($\pm$ 1.2) $\times 10^2$, and 1.1 ($\pm$ 0.1) $\times 10^3$ M$^{-1}$ respectively, indicating an enhanced binding force between cations and cavities of pillararenes, which was consistent with our hypothesis.

**P5-COOH**, the precursor of **WP5-M**, was almost insoluble in aqueous solution, and the ionization of its carboxyl hydrogen is incomplete. Notably, Ogoshi reported that the rotational barrier of **P5-COOH** in DMF-$d_6$ was 17.08 kcal/mol$^{-1}$ [49], which was lower than **WP5-Li** and **WP5-Na** but higher than **WP5-K**, **WP5-Rb** and **WP5-Cs**. This may be

**Table 1 | The summery of kinetic parameters of WP5-M in D$_2$O**

| Rotors[a] | $\Delta H^{\ddagger}$ (kcal mol$^{-1}$) | $\Delta S^{\ddagger}$ (cal mol$^{-1}$ K$^{-1}$) | $k_{298 K}$[b] (s$^{-1}$) | $\Delta G^{\ddagger}_{298 K}$ (kcal mol$^{-1}$) |
|---|---|---|---|---|
| **WP5-Li** | 29.83 | 38.92 | 0.26 | 18.23 |
| **WP5-Na** | 21.32 | 12.67 | 0.82 | 17.55 |
| **WP5-K** | 11.47 | −15.04 | 13.04 | 15.95 |
| **WP5-Rb** | 6.22 | −31.74 | 18.03 | 15.68 |
| **WP5-Cs** | 5.04 | −33.65 | 53.47 | 15.06 |

[a]The concentration of rotors was 10.0 mM.
[b]Rotational speed was revealed based on Eyring Plot.

ascribed to the steric favor to hydrogen than Li$^+$ and Na$^+$, although Li$^+$ and Na$^+$ had modest electronic favor in rotation process. For K$^+$, Rb$^+$ and Cs$^+$, the stabilizing effect brought by electronic effect would overcome the unfavorable steric hindrance. VT-NMR study on **P5-COOH** in a mixed solvent of DMF-$d_6$/H$_2$O (v/v = 3/1, Supplementary Figs. 24–25) was performed. The addition of water in organic solvent would facilitate ionization of carboxyl group (e.g., p$K_a$ value of acetic acid in water and DMSO were 4.76 and 12.60 respectively)[50], so the electronic effect during rotations of **P5-COOH** in the above mixed solvent could be amplified. A rotational barrier of 18.37 kcal/mol$^{-1}$ was revealed for **P5-COOH**, which was the highest energy barrier among **WP5** derivatives. This result suggested that if the carboxyl group could be fully ionized, the lubricant effect of the proton could be very weak owing to its smallest size compared with metal cations.

### Theoretical computations

Theoretical calculations were next carried out for validating our above-mentioned hypothesis. To begin with, the binding energy between cations and GS structures of **WP5** anion (pS conformation) was calculated ($E_{GS}$, Supplementary Table 1). $E_{GS}$ fell off in the sequence of NH$_4^+$ > Li$^+$ > Na$^+$ > K$^+$ (i.e., the stability sequence of GS: **WP5-NH$_4$** > **WP5-Li** > **WP5-Na** > **WP5-K**), which was inconsistent to experimental results. Hence, the difference among GS energy of **WP5-M** had minor effect on their discrepant rotational barriers. Then, we focused on deciphering transition states of **WP5-M** during rotations. Considering that the actual transition states of **WP5-M** may be highly distorted and hardly calculated[51], Stoddart's TS model was employed, where the transformation between pS and pR conformers of **WP5-M** could be dissected into four plausible pathways with serval possible intermediates (Fig. 5a and Supplementary Fig. 27)[33]. The potential energy surface scanning (PESS) was subsequently performed to monitor energy variation during rotations (Supplementary Fig. 28)[52,53]. The stage TS$_1$ was examined primarily (Fig. 5a), in which only one phenolic ring (marked in orange) was allowed free rotation around the cross-section of **WP5** (plane α, Fig. 5a). Due to the complexity of structures of **WP5** derivatives, it was difficult to locate transition state structures in rotation process. Instead, configurations with the highest energy sampled from PESS were considered as preliminary transition state structures for estimating rotational barriers (Supplementary Fig. 29)[54–56]. The calculation results were presented in Supplementary Figs. 30–33 and Supplementary Table 2, from which **WP5-Li** was found to have the highest rotational barrier of 8.83 kcal/mol for TS$_1$. Additionally, the binding energy between cations and TS$_1$ structures of **WP5** anion was also computed ($E_{TS}$, Supplementary Table 1). For the same cation, $E_{TS}$ was higher than its $E_{GS}$, which showcased stronger impact of cations on transition states than that on ground states.

To further understand rotation processes of **WP5-M**, the energy barrier for TS$_2$ was also evaluated. In this step, hydroquinone rings in ortho- (o-, TS$_2$a) or meta- (m-, TS$_2$b) position of initially rotated unit flipped around plane α (marked in blue, Fig. 5a). It was found that rotational barriers of TS$_2$a were higher than TS$_2$b in all studied systems (e.g., TS$_2$a: 16.18 kcal/mol; TS$_2$b: 13.84 kcal/mol for **WP5-Li**,

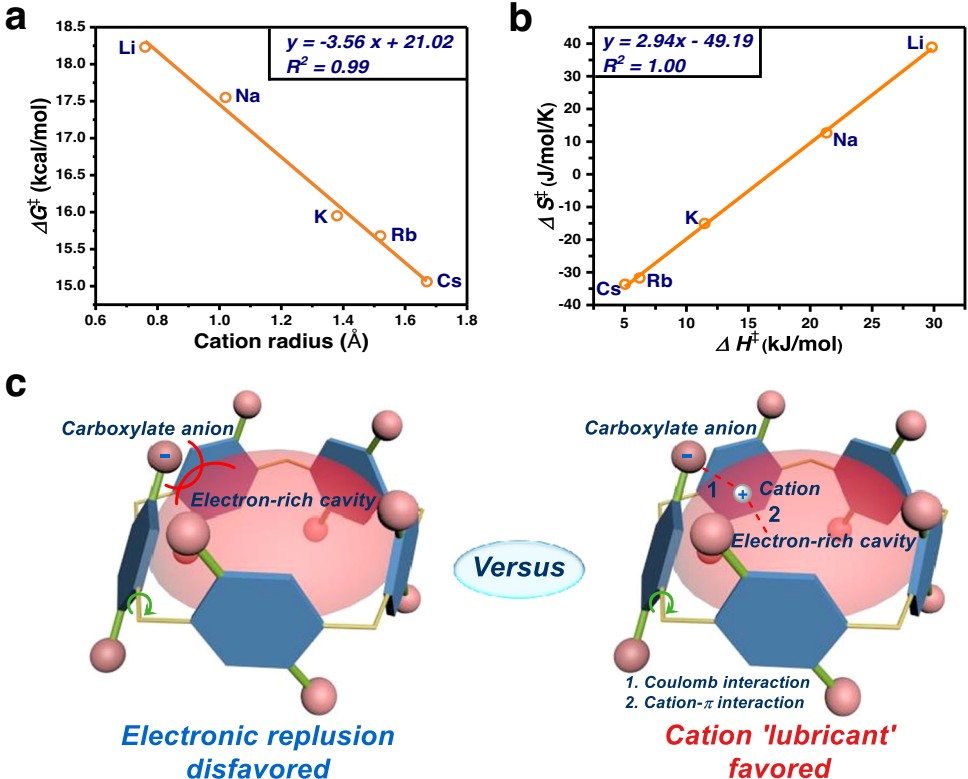

**Fig. 4 | Kinetic analysis of rotations in WP5-M. a** Correlation of the rotational barrier $\Delta G^{\ddagger}_{exp}$ with corresponding cation radius. **b** Enthalpy-entropy compensation plot of **WP5-M**. **c** Illustration of electron repulsion between carboxylate anion (represented with pink balls) and electron-rich cavity (represented with red translucent hemisphere) versus cation (represented with silver balls) lubricant effect.

Supplementary Figs. 30–33, Supplementary Table 2), which may be ascribed to steric hindrance of two neighboring hydroquinone rings in TS2a. Thus, TS2b was a favored pathway when the second hydroquinone ring tended to rotate. The highest energy barriers among two rotational processes (TS1 and TS2b) were evidently reduced from **WP5-Li**, **WP5-Na**, **WP5-K**, to **WP5-NH4**, which was in accordance with experimental results. Sampled structures from PESS were shown in Fig. 5b, in which hydroquinone rings rotated to the plane α for **WP5-Li**, **WP5-Na**, **WP5-K**, and **WP5-NH4**, respectively. A decrease in the nonbonded distance between cations and vertical phenolic rings in *m*-position (**WP5-Li**: 2.87 Å; **WP5-Na**: 2.74 Å; **WP5-K**: 2.35 Å; and **WP5-NH4**: 2.37 Å) indicated the enhanced cation-π interactions (Fig. 5b). Meanwhile, hydrogen bonding between ammonium ion and carboxylate oxygen of **WP5-NH4** was significant, which could be contributed to the lowest rotational barrier of **WP5-NH4** among these anionic pillar[5]arenes (Fig. 5b).

**Solvent effects**

Considering cations are highly solvated in water, and solvation might get involved in the rotation process of **WP5-M**, **WP5-Na** was chosen as the model to explore the effect of ionic solvation on rotations. We performed additional VT NMR studies in mixed solvent (D2O mixed with polar protic solvent methanol-$d_4$, or polar aprotic solvent DMSO-$d_6$), due to insolubility of **WP5-M** in nonaqueous solvent. Rotational barriers and relevant kinetic parameters of **WP5-Na** in the mixed solvent were equivalent to that in deuterium oxide (Supplementary Table 3 and Supplementary Figs. 34–43). This result indicated that the ionic solvation had little effect on rotation process, which may be ascribed to narrow cavities of **WP5** limited the solvation of cations in transition states. Since solvents were irrelevant to the rotational barriers of **WP5-M**, we attempted to study kinetic traits of **WP5-NH4** in a mixed solvent (D2O/ methanol-$d_4$, v/v = 3/2, the maximum mixing ratio

that be applied for dissolving **WP5-NH4** with a 10.0 mM concentration), of which the freezing point was about 228 K. Although a coalesce temperature of 268 K was revealed for **WP5-NH4**, unfortunately, the lower temperature couldn't lead the coalescent methylene peaks to fully split into two doublet (Supplementary Fig. 44). Hence, the rotational barrier of **WP5-NH4** could not be calculated according to current VT NMR data.

**Cation grease/brake rotations and fluorescence switch**

Having extensively explored the impact of cations on rotational barriers of **WP5**, attention was subsequently turned to construct switchable rotary motions in **WP5** platforms, where counter cations could grease/brake of rotors. $NH_4^+$ and $Li^+$ were chosen for an exchange pair, between which the switch could be easily achieved (Fig. 6a). Two doublets methylene peaks of **WP5-Li** were observed in $^1$H NMR spectra (Fig. 6b), showing slow rotations of rotors. NH4F was added for depositing $Li^+$ with $F^-$, leading to an acceleration of rotors (Fig. 6b). Subsequently, the solution was treated with LiOH, heating and bubbling with argon to remove $NH_3$, resulting in a recovery of slow rotations (Fig. 6b). More importantly, the switch of the rotational speed could recur with multiple cycles (Fig. 6c). Furthermore, an attempt of reversibly switching **WP5-NH4** and **WP5-Na** was also achieved, in which $Na^+$ could be captured with 15-crown-5, and $NH_4^+$ could be removed under heating and bubbling the solution with argon after basified by NaOH (Supplementary Figs. 45–46).

Encouraged by these findings, next we put our effort on modulating photochemical properties of **WP5**. Strikingly, a downward trend of fluorescence intensities of **WP5-M** under 365 nm irradiation was observed (Fig. 6d), which fall in line with the sequence of rotational barrier. **WP5-Li**, with the highest rotational barrier, had the strongest luminescence, while s quite weak fluorescence was observed for **WP5-NH4** (Fig. 6d). Presumably, restricted rotations

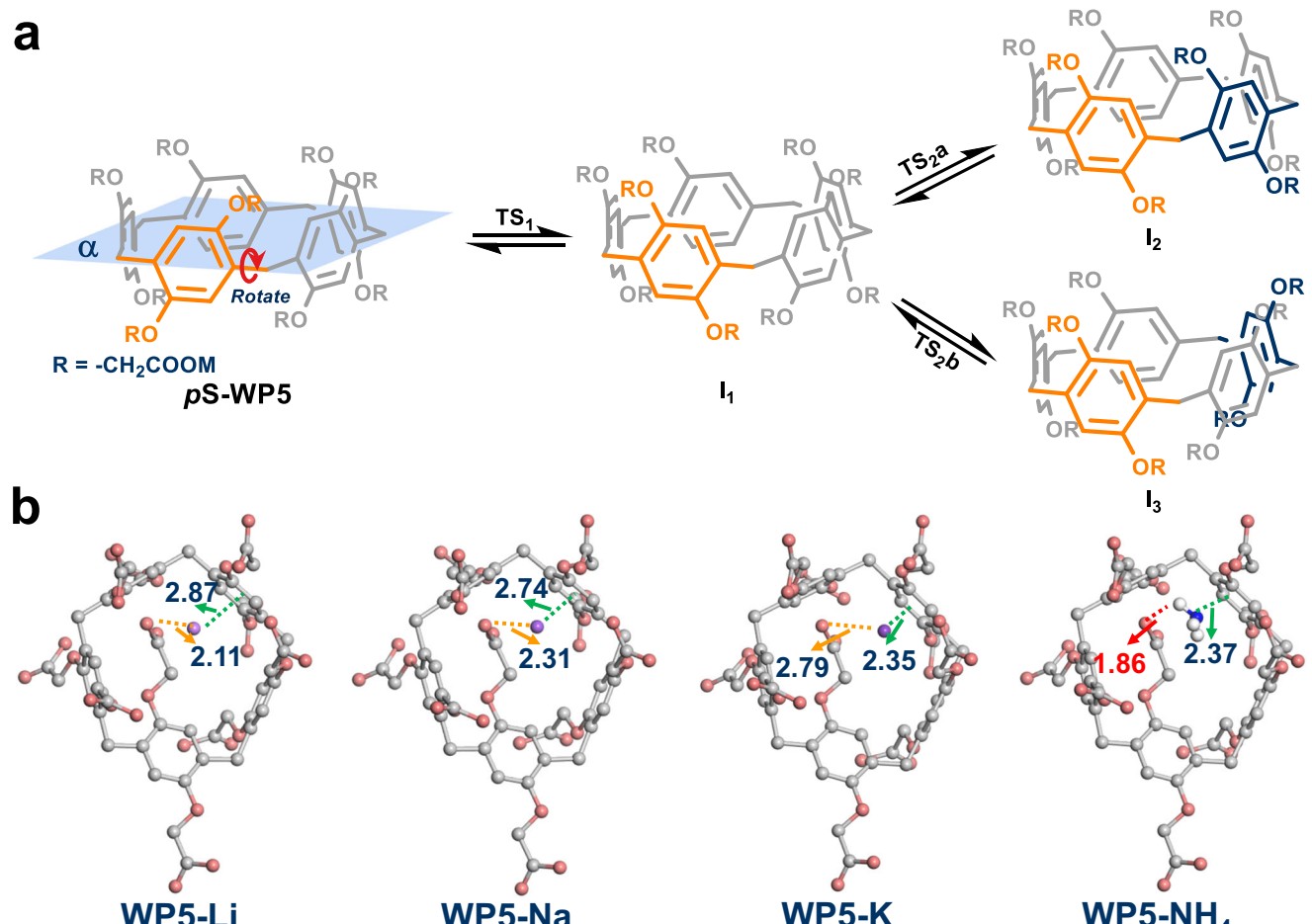

**Fig. 5 | Mechanism studies of rotation process of WP5-M. a** Partial pathways of transformation between its *p*S and *p*R conformers of **WP5-M. b** Configurations extracted from potential energy surface scanning when hydroquinone ring rotated to the plane α. The dashline represents the distance between cations and corresponding functional groups. The non-bonded distances are in units of Å.

reduced energy dissipations, and thus enhanced emission intensity, which was similar to RIR mechanisms of AIE phenomenon[11]. Furthermore, reversible fluorescence switching with several cycles was accomplished via the exchange of Li+ and NH4+ (Fig. 6e). This controllable fluorescence switch was promising to be considered in anti-counterfeiting technology (Fig. 6f). **WP5·NH4** aqueous solution was utilized as inks, and a number "7" could be written on filter paper, which was invisible under 365 nm UV irradiation (Fig. 6f (I)). When applying LiOH aqueous solution on to the surface of prewriting "7" image, the number could be revealed under 365 nm irradiation (Fig. 6f (II)). Moreover, the visible "7" image could be erased by painting NH4F solution onto the image surface again (Fig. 6f (III)).

## Discussion

In summary, the impact of cations on rotational barriers of anionic pillar[5]arenes has been investigated via dynamic NMR technology. It is found that cations had significant effects on rotational barriers of hydroquinone units in **WP5**, which is correlated with their radius. Next, theoretical computation is employed for providing mechanistic insights into rotation process. Due to orderly enhanced cation-π interactions between cations and electron-rich cavities in rotary processes, rotational barriers of **WP5-M** decrease from Li+ to Cs+. **WP5·NH4** has the lowest rotational barrier among **WP5-M**, which is ascribed to additional hydrogen bonding. Finally, switchable rotary motions of rotors are explored. The accelera-tion/deacceleration of rotors is

accomplished by means of cation exchange, and these findings further allow the fluorescence switch, which is applied as anti-counterfeiting inks. The strategies and results presented here should find potential applications in many fields, such as sensing, molecular devices, and smart materials.

## Methods

### Dynamic NMR studies

Rotational barriers of rotors **WP5-M** were measured via variable temperature (VT) [1]H NMR by monitoring the broadening of diastereotopic methylene protons in the rims of **WP5-M**. VT [1]H NMR spectra were recorded on a BRUKER AVANCE III 400 MHz or BRUKER AVANCE III 600 MHz spectrometer. The concentration of rotors were 10.0 mM. All VT experiments were performed three times. The NMR line shape analyses were performed on MestReNova 14 sofeware package (Mestrelab Research S. L.). Rates of exchange $k_{ex}$ were obtained by line width analysis with the equation:

$$k_{ex} = \pi(h - h_0) \tag{1}$$

where h represented the width of peak at half height, $h_0$ represented the width of peak at slow or no exchange[38].

Kinetic parameters were obtained by using the exchange rates ($k_{ex}$, s−1) obtained from line width analysis of the VT [1]H NMR spectra, the enthalpy change ($\Delta H^{\neq}$) and entropy change ($\Delta S^{\neq}$) of the transition

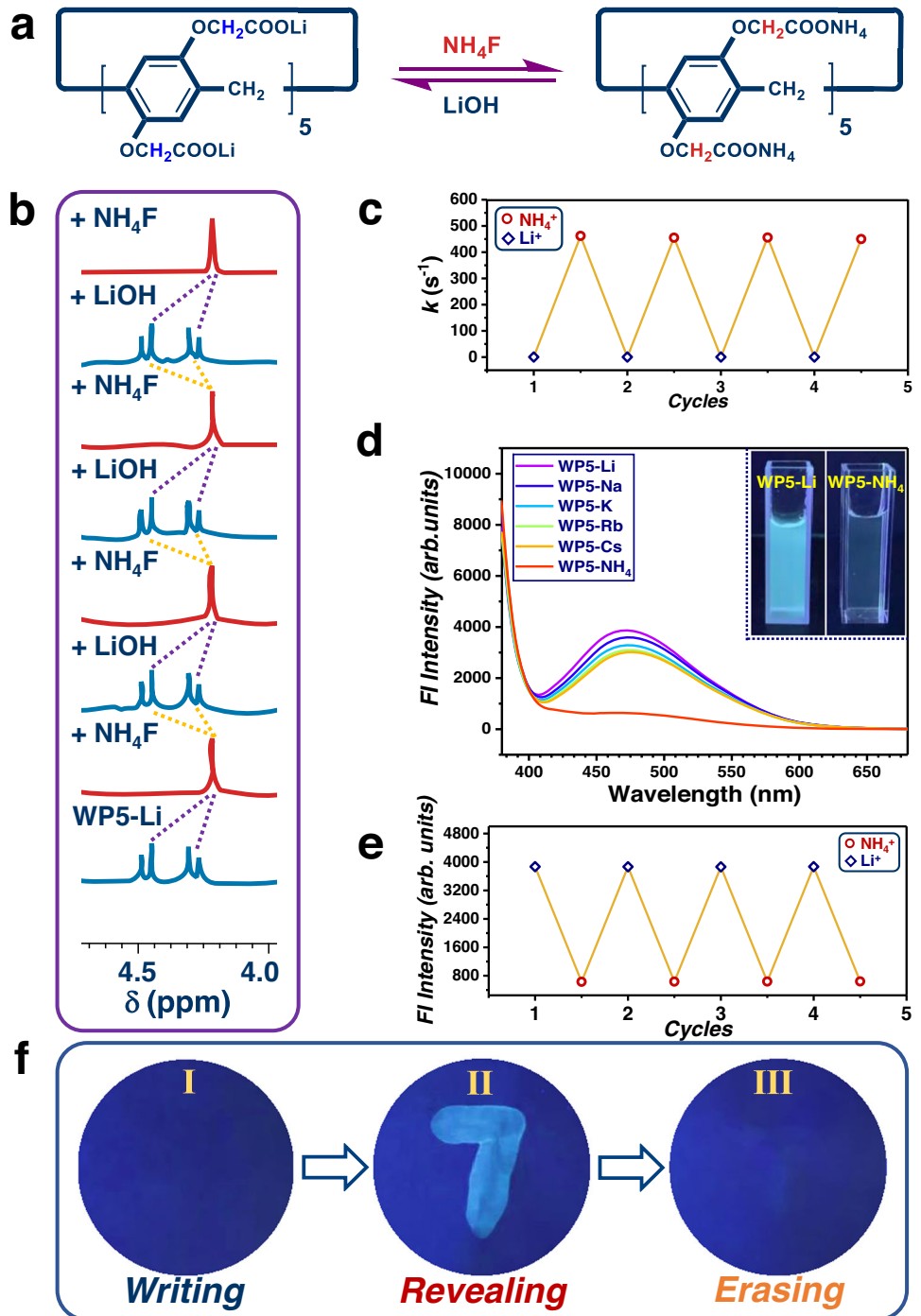

**Fig. 6 | Rotational speed control and fluorescence switch of WP5-M. a** Cation switch between Li⁺ and NH₄⁺. **b** Partial NMR of switching control of rotors. **c** Switching of rate for **WP5** in D₂O at 298 K with multiple cycles of cation exchange. **d** Fluorescence emission spectrum of **WP5-M** ($c = 5.0$ mM, $\lambda_{ex} = 365$ nm; insert: fluorescence of **WP5-Li** and **WP5-NH₄** under 365 nm UV irradiation). **e** Fluorescence response of **WP5** (5.0 mM) at 471 nm upon multiple cycles of cation switch between Li⁺ and NH₄⁺ in D₂O ($\lambda_{ex} = 365$ nm). **f** Revealing and erasing information with **WP5** inks: the image written with **WP5-NH₄** under 365 nm light (I), and the image could be observed upon adding LiOH on the pre-existing images (II), which could be erased by painting NH₄F (III).

state were calculated from Eyring plots:

$$\ln \frac{k}{T} = -\frac{\Delta H^{\ddagger}}{R}\frac{1}{T} + \frac{\Delta S^{\ddagger}}{R} + \ln\left(\frac{k_B}{h}\right) \qquad (2)$$

where k is the exchange rate constant, T is the absolute temperature, $\Delta H^{\ddagger}$ is the enthalpy of activation, $R$ is the universal gas constant, $k_B$ is the Boltzmann constant, $h$ is the Planck's constant, and $\Delta S^{\ddagger}$ is the

entropy of activation. The free energy of activation ($\Delta G^{\ddagger}$) was calculated through Gibbs equation:

$$\Delta G^{\ddagger} = \Delta H^{\ddagger} - T\Delta S^{\ddagger} \qquad (3)$$

The relevant NMR spectra and Eyring Plots were avalilble at the Supplementary Information.

## Theoretical calculation

All computational calculations were carried out with Gaussian 16 software[57]. The potential energy surface have been scanned using semi-empirical PM6 method with dispersion correction (PM6-D3)[52,53]. The sampled configurations from the scanned structures were further optimized using the B3LYP-D3 functional[58–61]. The standard 6−31 G(d) and 6−31 G + (d) basis sets were used for nonmetal atoms and metal Li, Na and K, respectively. For detail, please see Supplementary Information and Supplementary Data 1.

## Data availability

The authors declare that the all data supporting the findings of this study are available within this article and Supplementary Information files. The Supplementary Information file contains the experimental details and characterization of the compounds. Supplementary Data 1 contains the coordinates of the calculated structures. Supplementary Movie 1 is a demo of how rotations are controlled by cations in **WP5**.

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

## Acknowledgements

We gratefully thank the financial support of the National Natural Science Foundation of China (nos. 21871135, 21871136, 22071104, and 22033004), and the Starry Night Science Fund of Zhejiang University Shanghai Institute for Advanced Study (Grant No. SN-ZJU-SIAS-006). We are grateful to the High Performance Computing Centre of Nanjing University for providing the IBM Blade cluster system.

## Author contributions

H.Z. and L. F. contributed equally. H. Z. carried out the experimental work and analyzed the data. L. F. and J. M. performed theoretical computations. R. W., Y.S., C.S., and Y.C. synthesized the compounds. J.Jiao helped to conduct varible-temperature NMR test. H.Z., J.Jiang, and C.L. wrote the manuscript. L.W. revised the manuscript.

## Competing interests

The authors declare no competing interests.
