## [Peer Review File · Nature Communications]

Cation Controlled Rotation in Anionic Pillar[5]arenes and Its Application for Fluorescence SwitchREVIEWER COMMENTS

Reviewer #1 (Remarks to the Author):

This manuscript describes the rotational behavior of carboxylate pillar[5]arenes (WP5-M) with various counter-cations, and their application to fluorescence switch and anti-counterfeiting ink. The authors found that the rotational barriers of hydroquinone units in WP5 are significantly influenced by the counter-cations and succeeded in the construction of the cation-exchanging rotor system by using this counter-cation dependent rotation. The rotational barriers of various WP5-M were determined exactly through VT-NMR experiment followed by the line-shape simulation, and the rotational mechanism of WP5, which depends on counter-cations, was also rationally explained on the basis of the computational study. These are new insights in pillar[n]arene chemistry and very interesting. Furthermore, the strong luminescence of WP5-Li and the extremely weak fluorescence of WP5-NH₄ were revealed, and on the basis of these optical properties, the fluorescence switching system and anti-counterfeiting ink were achieved. Thus, this work is of significant interest to not only the organic chemistry community but also broad fields such as material science and nanotechnology. I judge that this paper meets the criteria for publication in Nature Communications. On the other hand, the following revisions would be required for the publication.

- 1) The authors did not evaluate the rotational barrier of WP5-NH₄ because of the limitation of the freezing point in D₂O. Meanwhile, it was revealed that the barriers are scarcely influenced by solvents. Did you conduct the VT-NMR experiment of WP5-NH₄ in the mixed solvent such as CD₃OD-D₂O possessing lower freezing point? I think that the rotational barrier of WP5-NH₄ is very important from the viewpoint of correlation with the emission intensity.
- 2) I think that the rotational barrier of WP5 (-COOH) should be added and compared with those of ionic species WP5-M.
- 3) I found a number of words containing "hyphen (-)" in the text (for example, line 67: synthe-sized, line 83: accu-racy, line 164: rota-tion, line 181: lumines-cence, line 209: cati-ons, line 303: Yamag-ishi). Please remove "hyphen".
- 4) Authors' name: Please change "C., L. I." to "Clarke, L. I.", (line 247), "S. R." to "Shiga, R." and "Yamagishi, T. A." to "Yamagishi, T.".
- 5) Line 108: Please change "radius" to "radii" or "were" to "was".
- 6) Supporting Information, page S4, ¹H NMR of WP5-K: Please correct "4.49 (...11H)" to 4.49 (...10H).

Reviewer #2 (Remarks to the Author):

The paper reports studies of anionic pillararene species which exhibit interconversion between two isomers through rotation of the hydroquinone subunits. This leads to a switching of the planar chirality of pillararene. It was found that this rotation is affected by the size of the counteraction, or more correctly, the strength of interaction between the cation and the pillararene core.

The observations are interesting and perhaps form the basis of a new type of molecular machine. However there are a number of questions which need to be addressed.

Firstly, the strength of the interaction between the cation and the pillararene host is important to the mechanism of the rotation. However, this does not appear to be quantified, except for a calculation. Is the strength of the interaction between non-anionic pillararene and the cations known. I appreciate that the cations would be expected to interact more favourably as they increase in size but this may be less clear to someone who doesn't work on pillararenes. Is there any evidence in the literature to back up this claim and if not the authors should measure the binding interaction between M(BF₄) (or equivalent) and a non-anionic pillararene, perhaps a fully alkylated version of WP5.

Figure 1 is a little unclear. Figure 1a and 1c are very similar except for the negative charge on the pink balls in part c. Presumably part a is also negatively charged so why the difference between the two parts?

The part on anti-counterfeiting seems like an unnecessary addition which adds little to this paper and would be better as a separate study.

Smaller points include:

Can the authors define AIE and AIEgens to make things simpler for the reader?

Throughout there are occasional hyphens which are unnecessary. And yet non-negligible should be hyphenated.

Overall, there is promise in the paper. I think it is interesting although I am unconvinced that this is of sufficient interest for the readership of Nature Communications, rather Communications Chemistry would be more appropriate.

Reviewer #3 (Remarks to the Author):

Hao Zheng and colleagues report an interesting study on the effect of cations on pillar[5]arene ring rotation with a nice correlation between cation radius and Gibbs free energy. The authors demonstrate that using water/D₂O as the solvent did not interfere with rotation thereby showing that it was the size of the cation and not its hydrate which caused the changes in speed of rotation. Though guest, solvents and substituents of different sizes have all been shown to affect pillar[5]arene pR to pS interconversion, to my knowledge this is the first report linking counterions to pillar[n]arene rotational kinetics.

The variable temperature NMR evidence supports different speeds of rotation. Based on the relative failure of a simple theoretical model, the authors calculated that it must be the rotational 'transition states' that are affected by the cations. This is entirely reasonable. The application of the 'on-off' fluorescence to write-erase-rewrite information is very compelling.

That noted, I am sure that the importance of this discovery warrants publication in Nature Communications. It is certainly interesting to those in the field but has very little that could be transferred to other macrocycles. Similarly, apart from the example given of disappearing ink, it is not clear how this discovery will drive the pillar[n]arene field forwards. If the Editor decides that it is suitable, the authors should note that the quality of the manuscript could be improved throughout. The following examples from the abstract alone illustrate this:

line 19

"manual manipulation of motions in molecule level" should read "manual manipulation of motion at the molecular level"

line 21

"it is urgent to develop novel rotor systems with controllable fashions" should read "the development of novel rotor systems, controlled by external stimuli, is urgently needed"

line 27

"their applications in fluorescence switch and anti-counterfeiting ink" should read "their applications as fluorescence-controlled switches and as anti-counterfeiting ink"

lines 28-29

The final sentence addresses the potential impact of the study and does not describe the work itself. It is therefore redundant and should be removed.

Other issues to address are:

line 32-33

The authors state that there are "various potential applications" of molecular rotors in different fields

but do not explain what these applications are. It would be useful for the reader to be told a little more to give the research some context.

line 55

In the review cited by the authors to support the statement that aggregation-induced emission arise from restricted rotational behaviour, only one of the 87 papers referenced notes that this may be the origin of changes observed in fluorescence. As its name suggests, AIE is caused by aggregation which, sometimes in the case of pillar[n]arenes, is due to guest inclusion and its reversal. Only when 'through-the-anulus-rotation' affects inclusion properties can this rotation truly be shown as the initiating factor in AIE.

Responses to comments from Reviewer 1:

“This manuscript describes the rotational behavior of carboxylate pillar[5]arenes (**WP5-M**) with various counter-cations, and their application to fluorescence switch and anti-counterfeiting ink. The authors found that the rotational barriers of hydroquinone units in **WP5** are significantly influenced by the counter-cations and succeeded in the construction of the cation-exchanging rotor system by using this counter-cation dependent rotation. The rotational barriers of various **WP5-M** were determined exactly through **VT-NMR** experiment followed by the line-shape simulation, and the rotational mechanism of **WP5**, which depends on counter-cations, was also rationally explained on the basis of the computational study. These are new insights in pillar[n]arene chemistry and very interesting. Furthermore, the strong luminescence of **WP5-Li** and the extremely weak fluorescence of **WP5-NH₄** were revealed, and on the basis of these optical properties, the fluorescence switching system and anti-counterfeiting ink were achieved. Thus, this work is of significant interest to not only the organic chemistry community but also broad fields such as material science and nanotechnology. I judge that this paper meets the criteria for publication in Nature Communications. On the other hand, the following revisions would be required for the publication.”

Q1: “The authors did not evaluate the rotational barrier of **WP5-NH₄** because of the limitation of the freezing point in D₂O. Meanwhile, it was revealed that the barriers are scarcely influenced by solvents. Did you conduct the VT-NMR experiment of **WP5-NH₄** in the mixed solvent such as CD₃OD-D₂O possessing lower freezing point? I think that the rotational barrier of **WP5-NH₄** is very important from the viewpoint of correlation with the emission intensity.”

Many thanks for your valuable suggestions. We have tried to conduct VT studies of **WP5-NH₄** in mixed solvent (D₂O/CD₃OD = 3:2, the maximum volume ratio of mixed solvent that could be applied for dissolving **WP5-NH₄** with a 10.0 mM concentration), of which the freezing point was about 228 K. Although a coalesce temperature of 268

K for **WP5-NH₄** was revealed, the lower temperature couldn't lead the coalescent methylene peaks to fully split into two doublet (Figure 1). Hence, the rotational barrier of **WP5-NH₄** could not be calculated based on current VT NMR data.

Figure 1. VT NMR spectra of **WP5-NH₄** in mixed solvent (D₂O/CD₃OD = 3:2).

We have added this result in the revised manuscript which was listed below:

“Since solvents were irrelevant to the rotational barriers of **WP5-M**, we attempted to study kinetic traits of **WP5-NH₄** in a mixed solvent (D₂O/methanol-*d*₄, v/v = 3/2, the maximum mixing ratio that be applied for dissolving **WP5-NH₄** with a 10.0 mM concentration), of which the freezing point was about 228 K. Although a coalesce temperature of 268 K was revealed for **WP5-NH₄**, unfortunately, the lower temperature couldn't lead the coalescent methylene peaks to fully split into two doublet (Supplementary Fig. 44). Hence, the rotational barrier of **WP5-NH₄** could not be calculated according to current VT NMR data.” (*please see page 10, line 194 in the revised manuscript*)

Q2: “I think that the rotational barrier of **WP5 (-COOH)** should be added and compared with those of ionic species **WP5-M**.”

Thanks a lot for your helpful comments. Since the carboxylic-substituted pillar[5]arene (**P5-COOH**, Figure 2) is almost insoluble in aqueous solution, and the ionization of its carboxyl hydrogen is incomplete, we didn't mention in the original manuscript. In this work, we focused on carboxylate pillar[5]arenes with various monovalent counterions (**WP5-M**), which could be completely ionized in water.

Ogoshi reported the rotational barrier of **P5-COOH** in DMF-*d*₆ was 17.08 kcal/mol¹ (*Chem. Commun.* **2010**, 46, 3708-3710), which was lower than **WP5-Li** and **WP5-Na** but higher than **WP5-K**, **WP5-Rb** and **WP5-Cs**. This may be ascribed to the steric favor to hydrogen than Li⁺ and Na⁺, despite Li⁺ and Na⁺ had modest electronic favor in rotation process. We performed VT-NMR study on **P5-COOH** in DMF-*d*₆/H₂O (v/v = 3/1, Figure 2 and Figure 3). Addition of water in organic solvent would facilitate ionization of carboxyl group (*e.g.*, p*K*_a value of acetic acid in water and DMSO were 4.76 and 12.60, respectively, *Acc. Chem. Res.* **1988**, 21, 456-463). The rotational barriers of **P5-COOH** in the above mixed solvent was 18.37 kcal/mol¹, which was the highest energy barrier among **WP5** derivatives. This result complied with our hypothesis, and it suggested that if the carboxyl group could be fully ionized, the ‘lubricant effect’ of the proton could be extremely weak compared with **WP5-M** species.

Figure 2. VT NMR spectra of **P5-COOH** in $\text{DMF-}d_6/\text{H}_2\text{O}$ ($v/v = 3/1$) between 253 K and 353 K. The coalesce temperature was 353 K.

Figure 3. Eyring plot of the rates of exchange in $\text{DMF-}d_6/\text{H}_2\text{O}$ ($v/v = 3/1$) obtained from line width analysis of methene protons on **P5-COOH**. The barrier ($\Delta G^\ddagger = 18.37 \text{ kcal}\cdot\text{mol}^{-1}$) was calculated from the slope ($\Delta H^\ddagger = 24.24 \text{ kcal}\cdot\text{mol}^{-1}$) and y-intercept ($\Delta S^\ddagger = 19.70 \text{ cal}\cdot\text{mol}^{-1}\cdot\text{K}^{-1}$).

We have added relevant discussion about rotations of **P5-COOH** in revised manuscript which was listed below:

“To explore the impact of counterions on rotations of hydroquinone units in anionic pillar[5]arenes, carboxylate pillar[5]arenes with various monovalent counterions were chosen (**WP5-M**, Fig. 1a), which could be completely ionized in water.”

“**P5-COOH**, the precursor of **WP5-M**, was almost insoluble in aqueous solution, and the ionization of its carboxyl hydrogen is incomplete. Notably, Ogoshi reported that the rotational barrier of **P5-COOH** in DMF-*d*₆ was 17.08 kcal/mol⁻¹,⁴⁹ which was lower than **WP5-Li** and **WP5-Na** but higher than **WP5-K**, **WP5-Rb**, and **WP5-Cs**. This may be ascribed to the steric favor to hydrogen than Li⁺ and Na⁺, although Li⁺ and Na⁺ had modest electronic favor in rotation process. For K⁺, Rb⁺, and Cs⁺, the stabilizing effect brought by electronic effect would overcome the unfavourable steric hindrance. VT-NMR study on **P5-COOH** in a mixed solvent of DMF-*d*₆/H₂O (v/v = 3/1, Supplementary Fig. 24 - 25) was performed. The addition of water in organic solvent would facilitate ionization of carboxyl group (*e.g.*, p*K*_a value of acetic acid in water and DMSO were 4.76 and 12.60, respectively),⁵⁰ so the electronic effect during rotations of **P5-COOH** in the above mixed solvent could be amplified. A rotational barrier of 18.37 kcal/mol⁻¹ was revealed for **P5-COOH**, which was the highest energy barrier among **WP5** derivatives. This result suggested that if the carboxyl group could be fully ionized, the ‘lubricant effect’ of the proton could be extremely weak owing to its smallest size compared with metal cations.” (*please see page 3, line 69 and page 7, line 127 in the revised manuscript*)

Q3: “I found a number of words containing “hyphen (-)” in the text (for example, line 67: synthe-sized, line 83: accu-racy, line 164: rota-tion, line 181: lumines-cence, line 209: cati-ons, line 303: Yamag-ishi). Please remove “hyphen”.”

Thank you for your corrections. We have removed redundant “hyphen” in our revised manuscript.

Q4: “Authors’ name: Please change “C., L. I.” to “Clarke, L. I.”, (line 247), “S. R.” to “Shiga, R.” and “Yamagishi, T. A.” to “Yamagishi, T.”.”

We have revised the author's abbreviation format in our manuscript. Thank you for your correction.

Q5: “Line 108:

Please change “radius” to “radii” or “were” to “was”.”

We have changed “radius” to “radii” in our revised manuscript, and thank you for your correction.

Q6: “Supporting Information, page S4, ¹H NMR of WP5-K: Please correct “4.49 (...11H)” to 4.49 (...10H).”

Thanks for your kindly correction. We have corrected relevant NMR description.

Responses to comments from Reviewer 2:

“The paper reports studies of anionic pillararene species which exhibit interconversion between two isomers through rotation of the hydroquinone subunits. This leads to a switching of the planar chirality of pillararene. It was found that this rotation is affected by the size of the counteraction, or more correctly, the strength of interaction between the cation and the pillararene core. The observations are interesting and perhaps form the basis of a new type of molecular machine. However, there are a number of questions which need to be addressed.”

Q1: “Firstly, the strength of the interaction between the cation and the pillararene host is important to the mechanism of the rotation. However, this does not appear to be quantified, except for a calculation. Is the strength of the interaction between non-anionic pillararene and the cations known. I appreciate that the cations would be

expected to interact more favorably as they increase in size but this may be less clear to someone who doesn't work on pillararenes. Is there any evidence in the literature to back up this claim and if not the authors should measure the binding interaction between M(BF₄) (or equivalent) and a non-anionic pillararene, perhaps a fully alkylated version of WP5.”

Many thanks for your valuable suggestion. We made the hypothesis that the strength between cations and cavities of **WP5** would increase with the cation size increasing, which was based on a size-specific mechanism for interaction of alkali metal ions with crown ether (*Chem. Rev.* **2004**, 104, 2723-2750). The binding constant of 18-crown-6 with K⁺ was higher than that with Na⁺, because the size of K⁺ was more matched with the cavity of 18-crown-6. Actually, this size-specific mechanism is ubiquitous in macrocyclic host-guest systems (*Coord. Chem. Rev.* **2022**, 467, 214580; *Chem. Rev.* **2022**, 122, 9032-9077; *Chem. Rev.* **2019**, 119, 9753-9835.). For solidifying our conclusion, we measured binding constants of M⁺ with ethoxycarbonylmethoxy-substituted pillar[5]arenes (**P5-COOEt**) by UV titration (Figure 4, Figure 5 left). Addition of MBF₄ aqueous solution to a DMF/H₂O solution (v/v = 4/1) with the same concentration (0.10 mM) of **P5-COOEt** resulted in an increase of the intensity of the CT band of the complex.

Figure 4. UV titration for measurement of binding constant between **P5-COOEt** and cations.

The mole ratio plots for the complexation between **P5-COOEt** and cations indicated a 1:1 stoichiometry (Figure 5 mid). Treatment of the collected absorbance data with a non-linear curve-fitting program afforded the corresponding association constant (K_a).

The non-linear curve-fitting was based on the equation (K. A. Connors, *Binding Constants*, Wiley: New York, **1987**; *J. Am. Chem. Soc.* **1996**, *118*, 4931-4951):

$$\Delta A = (A_{\infty}/[H]_0) (0.5[G]_0 + 0.5 ([H]_0 + 1/K_a) - (0.5 (([G]_0)^2 + (2[G]_0(1/K_a - [H]_0)) + (1/K_a + [G]_0^2)^{0.5})),$$

where ΔA is the change of absorption intensity of **P5-COOEt** at 296 nm after addition of MBF_4 solution, A_{∞} is the absorption intensity of the charge-transfer band when the host is completely complexed, $[H]_0$ is the fixed initial concentration of the host, and $[G]_0$ is the varying concentration of the guest.

Figure 5. Titration curve (left), mole ratio plot (mid) and non-linear fitting curve (right) of host **P5-COOEt** and guest K^+ , Rb^+ , and NH_4^+ in DMF/ H_2O solution (v/v = 4/1).

By the non-linear curve-fitting methods, a binding constant K_a was estimated to be $4.2 (\pm 1.3) \times 10^2$, $7.9 (\pm 1.2) \times 10^2$, $1.1 (\pm 0.1) \times 10^3 \text{ M}^{-1}$ for K^+ , Rb^+ , and NH_4^+ respectively, which was echoed with barrier data. It was failed to obtain K_a of Li^+

and Na⁺, due to the fact that the changes of UV absorption were too small to conduct the non-linear curve-fitting.

We have added the relevant discussion about UV titration of **P5-COOEt** with cations in revised manuscript which was listed below:

“Ascribing to a size-specific mechanism that was ubiquitous in macrocyclic host-guest systems (*e.g.*, the binding constant of 18-crown-6 with K⁺ was higher than that with Na⁺ and the size of K⁺ that was more matched with the cavity of 18-crown-6),⁴⁴⁻⁴⁶ larger counter cations might have stronger interactions, leading to an enhanced ‘lubrication effect’ (Fig. 4c).”

“To preliminarily validate this size-specific mechanism in pillar[5]arenes, we measured binding constants (K_a) of K⁺, Rb⁺, and NH₄⁺ with ethoxycarbonylmethoxy-substituted pillar[5]arene (**P5-COOEt**, Fig. 2) in DMF/H₂O solution (v/v = 4/1) using titration method (Supplementary Fig. 26).⁴⁷⁻⁴⁸ Although it failed to measure K_a of Li⁺ and Na⁺, due to the fact that the changes of their UV absorption were too small to conduct the non-linear curve-fitting, an increased K_a of K⁺, Rb⁺, and NH₄⁺ was successfully estimated to be $4.2 (\pm 1.3) \times 10^2$, $7.9 (\pm 1.2) \times 10^2$, and $1.1 (\pm 0.1) \times 10^3$ M⁻¹ respectively, indicating an enhanced binding force between cations and cavities of pillararenes, which was consistent with our hypothesis.” (*please see page 6, line 113 and page 7, line 119 in the revised manuscript*)

Q2: “Figure 1 is a little unclear. Figure 1a and 1c are very similar except for the negative charge on the pink balls in part c. Presumably part a is also negatively charged so why the difference between the two parts?”

Thanks a lot for your helpful comments. It is our negligence to make the two illustrations inconsistent. We revised Figure 1 in the revised manuscript to make it clearer, which was listed below:

“**Fig. 1** Overview of controllable rotor platforms based on **WP5** skeletons.

a Illustration of planar chirality transformation of **WP5-M**. **b** ^1H NMR spectra of **WP5-M** in D_2O (10.0 mM, 298 K). Peaks of methylene groups are shown in green. **c** Cartoon Illustrations of cation-modulated switchable rotary motion of **WP5-M**. For simplification, only one rotary phenolic ring of **WP5** is shown.” (please see page 4, line 78 in the revised manuscript)

Q3: “The part on anti-counterfeiting seems like an unnecessary addition which adds little to this paper and would be better as a separate study.”

Thank you for your helpful suggestions. The anti-counterfeiting ink is one of promising applications of cation grease/brake **WP5** rotor platform, showing the preliminary progress from the theoretical studies to functional systems. We expect to develop further applications which are based on this preliminary exploration, and relevant works are in progress. Besides, as the reviewer 3 said “*The application of the ‘on-off’ fluorescence to write-erase-rewrite information is very compelling.*”, the anti-counterfeiting technology is one of highlights of our work, and hence, it would be better to keep this part in the manuscript.

Q4: “Smaller points include: can the authors define AIE and AIEgens to make things simpler for the reader?”

Thanks for your helpful comments. We have added relevant description about AIE and AIEgens in the revised manuscript listed below:

“In addition, it is worth noting that one of the working mechanisms of the aggregation-induced emission (AIE) phenomena was the restriction of intramolecular rotations (RIR, *e.g.*, tetraphenylethylene derivatives), of which the aggregation could reduce the speed of intramolecular rotations, lowering energy dissipations, and thus the fluorescence intensity could be enhanced.¹¹ Moreover, pillar[n]arenes have been recently reported as novel AIEgens (luminogens exhibiting AIE attributes), and their AIE properties were arisen from restricted rotations of their phenolic rings.³⁹⁻⁴⁰ Inspired by that, we inferred controlling rotary speed of hydroquinone rings in **WP5** would thus result in fluorescence switch.” (*please see page 3, line 57 in the revised manuscript*)

Q5: “Throughout there are occasional hyphens which are unnecessary. And yet non-negligible should be hyphenated.”

Thank you for your corrections. We have removed redundant “hyphen” in our revised manuscript.

Q6: “Overall, there is promise in the paper. I think it is interesting although I am unconvinced that this is of sufficient interest for the readership of Nature Communications, rather Communications Chemistry would be more appropriate.”

We quite understand your concerns, the major scientific ideas and merits of our work might not be fully described because our expressions are probably not good enough. The presented results in our manuscript are novel and significant for the following reasons: 1. The development of new controllable rotor platforms is highly desired; 2. This work discovers the non-negligible effect of counter cations on the planar chirality transformation of anionic pillar[5]arenes for the first time, and it is also the first

example of establishing tunable and reversible rotary motions in pillar[n]arenes derivatives; 3. Additionally, it is the first time to regulate fluorescence emission of **WP5** by cation switch; 4. The strategy exhibited here will be appealing to future research of sensing, molecular devices, and smart materials. We believe that this work is appropriate to the readership of *Nature Communications* not only for the scientific value but also for its potential applications.

Responses to comments from Reviewer 3:

“Hao Zheng and colleagues report an interesting study on the effect of cations on pillar[5]arene ring rotation with a nice correlation between cation radius and Gibbs free energy. The authors demonstrate that using water/D₂O as the solvent did not interfere with rotation thereby showing that it was the size of the cation and not its hydrate which caused the changes in speed of rotation. Though guest, solvents and substituents of different sizes have all been shown to affect pillar[5]arene pR to pS interconversion, to my knowledge this is the first report linking counterions to pillar[n]arene rotational kinetics.

The variable temperature NMR evidence supports different speeds of rotation. Based on the relative failure of a simple theoretical model, the authors calculated that it must be the rotational ‘transition states’ that are affected by the cations. This is entirely reasonable. The application of the ‘on-off’ fluorescence to write-erase-rewrite information is very compelling.”

Q1: “That noted, I am sure that the importance of this discovery warrants publication in *Nature Communications*. It is certainly interesting to those in the field but has very little that could be transferred to other macrocycles. Similarly, apart from the example given of disappearing ink, it is not clear how this discovery will drive the pillar[n]arene field forwards. If the Editor decides that it is suitable, the authors should note that the quality of the manuscript could be improved throughout. The following examples from the abstract alone illustrate this:”

Thank you for your helpful comments. Our goal is to develop novel molecular machines based on macrocyclic skeletons. Modulation on the rotational speed of phenolic ring in pillar[5]arene is the first step, and we expect that the rotational direction of these benzene ring could be controlled later. Besides, the strategy exhibited in our research is not only useful for pillar[n]arene chemistry, but also for other macrocycles with flipping aromatic walls, such as calix[n]arene, zorb[n]arenes and oxatub[n]arenes *etc.* (*Chem. Soc. Rev.* **2020**, *49*, 4176-4188). We believe our work is suitable for publication in *Nature Communications*, not only for the scientific value but also for its potential applications in macrocyclic chemistry, molecular devices, and smart materials.

Q2: “ line 19: “manual manipulation of motions in molecule level” should read “manual manipulation of motion at the molecular level”; line 21: “it is urgent to develop novel rotor systems with controllable fashions” should read “the development of novel rotor systems, controlled by external stimuli, is urgently needed”; line 27: “their applications in fluorescence switch and anti-counterfeiting ink” should read “their applications as fluorescence-controlled switches and as anti-counterfeiting ink”; lines 28-29: The final sentence addresses the potential impact of the study and does not describe the work itself. It is therefore redundant and should be removed.”

Thanks a lot for your helpful comments. We have revised relevant sentences in our manuscript which was listed below:

“The manual manipulation of motions at the molecule level has always been a hot topic in the field of chemistry. Molecular rotors have wide applications in building nanomachines and functional materials, due to their controllable rotations. Hence, the development of novel rotor systems, controlled by external stimuli, is urgently needed. Differential kinetic traits of planar chirality transformation in sodium carboxylate pillar[5]arene (**WP5-Na**) and ammonium carboxylate pillar[5]arene (**WP5-NH₄**) were observed serendipitously, which inspired us to construct a promising rotary platform in anionic pillar[5]arenes (**WP5**) skeletons. Herein, we demonstrate for the first time the non-negligible effect of counter cations on rotational barriers of hydroquinone rings in

WP5, which further exhibit its utilities in establishing a novel and tunable rotor system. Finally, cation grease/brake rotations of **WP5** and their applications as fluorescence switch and anti-counterfeiting ink were explored.” (*please see page 1, line 19 in the revised manuscript*)

Q3: “Other issues to address are: line 32-33: The authors state that there are “various potential applications” of molecular rotors in different fields but do not explain what these applications are. It would be useful for the reader to be told a little more to give the research some context.”

Thank you for your helpful suggestions. We supplemented the description of two applications of controllable molecular rotors in the revised manuscript which was listed below:

“Molecular rotors have various potential applications in molecular devices,¹⁻³ medicine,⁴⁻⁶ asymmetric catalysis,⁷⁻⁸ and smart materials,⁹⁻¹¹ which were derived from their controllable rotations. For example, light-driven molecular motors could drill through cell membranes using their molecular-scale actuation, inducing necrosis and facilitating chemical species into live cells.⁵ Feringa and co-workers developed a novel artificial muscle-like functional materials which was based on supramolecular assembly of photo-responsive molecular motors.⁹” (*please see page 2, line 31 in the revised manuscript*)

Q4: “line 55: In the review cited by the authors to support the statement that aggregation-induced emission arise from restricted rotational behavior, only one of the 87 papers referenced notes that this may be the origin of changes observed in fluorescence. As its name suggests, AIE is caused by aggregation which, sometimes in the case of pillar[n]arenes, is due to guest inclusion and its reversal. Only when ‘through-the-anulus-rotation’ affects inclusion properties can this rotation truly be shown as the initiating factor in AIE.”

The AIE property of pillar[n]arene reported by Chen (*J. Mater. Chem. C*, **2019**, 7,11747-11751) is just an inspiration that made us to regulate fluorescence emission of **WP5-M** by modulating their rotational barriers. One of the working mechanisms of the AIE phenomena was the restriction of intramolecular rotations, such as HPS or TPE derivatives (Figure 6, *Chem. Rev.* **2015**, 115, 11718-11940). The aggregation of these AIE molecules could reduce the speed of intramolecular rotations, lowering energy dissipations, and thus the fluorescence intensity could be enhanced. It is reasonable to consider that controlling rotary speed of hydroquinone rings in **WP5** by cation switch would result in modulation of fluorescence intensities. It is our negligence to make some misunderstanding to readers about this part.

Figure 6. Structures of AIEgens HPS and TPE.

We have revised related descriptions in the latest manuscript which was listed below:

“In addition, it is worth noting that one of the working mechanisms of the aggregation-induced emission (AIE) phenomena was the restriction of intramolecular rotations (RIR, *e.g.*, tetraphenylethylene derivatives), of which the aggregation could reduce the speed of intramolecular rotations, lowering energy dissipations, and thus the fluorescence intensity could be enhanced.¹¹ Moreover, pillar[n]arenes have been recently reported as novel AIEgens (luminogens exhibiting AIE attributes), and their AIE properties were arisen from restricted rotations of their phenolic rings.³⁹⁻⁴⁰ Inspired by that, we inferred controlling rotary speed of hydroquinone rings in **WP5** would thus result in fluorescence switch.” (*please see page 3, line 57 in the revised manuscript*)

REVIEWERS' COMMENTS

Reviewer #1 (Remarks to the Author):

I reviewed a previous submission of this work, and the authors have satisfactorily addressed the concerns that I raised therein. I strongly recommend the publication after correction of the following careless mistakes.

1. page 9, line 168: Please change "Fig. 4a" to "Fig. 5a".
2. Page 15, line 295, ref.8: Please change "Grill, K.;" to "Grill, K.,".

Reviewer #2 (Remarks to the Author):

The authors have addressed the comments of the reviewers, providing robust scientific responses. In this sense the paper is now ready for publication. I am still unsure that the manuscript is at a level of significance that merits publication in Nature Communications but I respect the editorial decision in this regard.

Responses to comments from Reviewer 1:

“I reviewed a previous submission of this work, and the authors have satisfactorily addressed the concerns that I raised therein. I strongly recommend the publication after correction of the following careless mistakes.”

Thank you for your helpful comments for improving our manuscript and convey you our sincere regards!

Q1: “1. page 9, line 168: Please change "Fig. 4a" to "Fig. 5a".”

Thank you for your corrections. We have changed ‘Fig. 4a’ to ‘Fig. 5a’ in our revised manuscript.

Q2: “2. Page 15, line 295, ref.8: Please change "Grill, K.;" to "Grill, K.,".”

Thank you for your corrections. We have changed ‘Grill, K.;

’ to ‘Grill, K.,’ in our revised manuscript.

Responses to comments from Reviewer 2:

“The authors have addressed the comments of the reviewers, providing robust scientific responses. In this sense the paper is now ready for publication. I am still unsure that the manuscript is at a level of significance that merits publication in Nature Communications but I respect the editorial decision in this regard.”

Thank you for your efficient work on reviewing our manuscript and convey you our sincere regards!